# Asbestos-Related Lung Cancer: A Hospital-Based Case-Control Study in Indonesia

**DOI:** 10.3390/ijerph17020591

**Published:** 2020-01-16

**Authors:** Anna Suraya, Dennis Nowak, Astrid Widajati Sulistomo, Aziza Ghanie Icksan, Elisna Syahruddin, Ursula Berger, Stephan Bose-O’Reilly

**Affiliations:** 1CIHLMU Center for International Health, University Hospital, LMU Munich, 80539 Munich, Germany; dennis.nowak@med.uni-muenchen.de; 2Occupational Health and Safety Program, Universitas Binawan, Jakarta 13630, Indonesia; 3Institute and Clinic for Occupational, Social and Environmental Medicine, University Hospital Munich (LMU), 80336 Munich, Germany; stephan.boeseoreilly@med.uni-muenchen.de; 4Member, German Center for Lung Research (DZL), Comprehensive Pneumology Center Munich (CPC-M), 81377 Munich, Germany; 5Occupational Medicine Specialist, Universitas Indonesia Academic Hospital, Jakarta 16424, Indonesia; asulistomo@gmail.com; 6Department of Radiology, Faculty of Medicine UPN Veteran, Persahabatan Hospital, National Respiratory Referral Hospital, Jakarta 13230, Indonesia; azizagicksan@yahoo.com; 7Division of Thoracic Oncology Department of Pulmonology Faculty of Medicine, Universitas Indonesia Persahabatan hospital, Jakarta 13230, Indonesia; elisna2002@gmail.com; 8IBE—Institute for Medical Information Processing, Biometry and Epidemiology, Ludwig Maximilian Universität, 81377 Munich, Germany; berger@ibe.med.uni-muenchen.de; 9Institute of Public Health, Medical Decision Making and Health Technology Assessment, Department of Public Health, Health Services Research and Health Technology Assessment, UMIT—Private University for Health Sciences, Medical Informatics and Technology, 6060 Hall in Tirol, Austria

**Keywords:** lung cancer, asbestos, Indonesia

## Abstract

Indonesia has limited data on asbestos-related diseases despite abundant use. This study investigated the risk of occupational asbestos exposure for lung cancer development, utilizing a hospital-based case-control study. Subjects were patients who received a thoracic CT scan at Persahabatan Hospital, Jakarta. The cases had primary lung cancer confirmed by histology, the controls were negative for lung cancer. The cumulative occupational asbestos exposure was calculated by multiplying the exposure intensity by the years of exposure. The exposure intensity was obtained by adopting the weighted arithmetic mean value of asbestos exposure from a job-exposure matrix developed in Korea. The primary data analysis was based on logistic regression. The study included 696 subjects, with 336 cases and 360 controls. The chance of lung cancer for subjects exposed to asbestos was doubled (OR = 2.04, 95% CI = 1.21–3.42) compared with unexposed, and subjects with a cumulative asbestos exposure of 10 fiber-years or more even showed an OR of 3.08 (95% CI = 1.01–9.46). The OR of the combined effect between smoking and asbestos was 8.7 (95% CI = 1.71–44.39); the interaction was consistent with an additive and multiplicative risk model. Asbestos exposure is associated with a higher chance of lung cancer. Improved policies are needed to protect the population from asbestos hazards.

## 1. Introduction

Lung cancer is the most common cancer in Indonesian men and the third most common in women, causing a total of 26,095 fatalities in 2018 [1]. With approximately 63% of Indonesian males smoking, it is an attractive risk factor for a lung cancer investigation [2]. However, studies have demonstrated that other risk factors such as asbestos, radon and diesel engine particulates, increase the risk of lung cancer [3]. It has been estimated that occupational lung cancer is responsible for 17–29% of all lung cancer deaths in men and asbestos is the most significant occupational carcinogen accounting for around 55–85% of all occupational lung cancer cases [4].

Asbestos is a general term for a family of fibrous silicate minerals with a crystalline structure that mesmerized the industrial world in the 1900s because of its unique chemical and physical characteristics [5]. Asbestos is strong, flexible, stable, durable and resistant to heat, chemical and biological degradation. For over a century it was used in more than 3000 different products, including construction materials, brake pads, insulation, textiles, plastic, paper and many others [6,7]. Health consequences related to asbestos arose following its extensive use. The first suspected asbestos-related lung cancers were reported by Gloyne in 1935 upon autopsies of two female asbestos textile factory workers who were found to suffer from asbestosis and lung cancer [8]. More cases of asbestos-related cancers have been diagnosed since that time and since 1987 asbestos has been classified by the International Agency for Research on Cancer of the World Health Organization as being carcinogenic to humans [9,10].

Indonesia started importing asbestos in 1950 and recently it joined the top five asbestos-importing countries in the world [11,12]. In the past ten years, more than one million tons of asbestos have been brought into Indonesia, mainly as raw material for roofing, cement, ceilings, partitions and in lesser amounts for brake systems, insulation and heat-resistant textiles [13]. Consumption of asbestos increased from 20,000 tons in the 1980s to 50,000 tons in the 1990s and continued to increase to roughly 150,000 tons in the 2000s. Based on the amount of asbestos processed, in 2017 the WHO estimated that there should be approximately 1000 lung cancer cases and 400 mesothelioma cases in Indonesia in that year [14]. In reality, until 2019, there were no confirmed reports of asbestos-related lung cancer or mesothelioma. In addition, occupational and environmental asbestos exposure data is very limited in the country.

Considering the extensive use of asbestos in Indonesia, an investigation of its health effects is necessary. This study is the first case-control study in Indonesia comparing occupational asbestos exposure to the risk of lung cancer development and the combined effect with smoking. This study is also the first to adopt the asbestos job exposure matrix (JEM) from a neighboring country: the general population job exposure matrix (GPJEM) from the Republic of South Korea. It was developed by Choi et al. [15] to quantify the amount of asbestos exposure in various job classifications in that country. We have adopted it due to the lack of data on workplace asbestos exposure for Indonesia. Improved scientific information about asbestos and its health effects may facilitate the development of asbestos management policies in Indonesia.

## 2. Materials and Methods

### 2.1. Study Population

This study was conducted at the National Respiratory Hospital, a 585 bed public hospital in Jakarta, Indonesia [16]. A case-control study was performed to identify the association between asbestos exposure and the risk of developing lung cancer. Inpatients and outpatients aged 35 years or older who had received a thoracic CT scan at the radiology department between May 2018 and August 2019 were recruited. The reasons for a thoracic CT scan were lung infection, mediastinal mass, lung nodule or mass, trauma, evaluation of pleural diseases and malignancies and other conditions.

The cases had primary lung cancer confirmed by histology regardless of the type of cancer. The controls were negative for lung cancer selected from the same group of patients receiving thoracic CT. Patients were excluded as controls if they had one of these conditions: mesothelioma, pleural plaques, asbestosis and interstitial lung diseases.

### 2.2. Assignment of Exposure

An occupational medicine physician evaluated the occupational and non-occupational asbestos exposures blind to the patient’s grouping as a case or control.

Occupational asbestos exposure was defined as a history of occupational contact with asbestos fibers at least ten years prior to the time of the interview. The description of occupational asbestos exposure in the workplace was obtained from a questionnaire adapted from Cancer Research UK [17]. It included industry category, type of job, time of first contact, duration of work, the number of work hours in a day and the number of workdays per week. Subjects’ occupations were coded according to five-digit Indonesian Standard Industrial Classification (ISIC) 2015.

The cumulative occupational asbestos exposure was expressed in fiber-years, which was fiber concentration in weighted arithmetic mean (WAM), expressed in fiber/milliliter (f/mL) of air multiplied by the total duration of exposure in years. A year was defined as exposure during 8 h shift over 240 workdays and spread over 48 weeks. Fiber concentration in WAM was obtained from the Korean GPJEM by linking the five-digit ISIC 2015 to the Korean Standard Industrial Classification (KSIC, 9th edition) and Korean Standards Classification of Occupation code (6th edition) to the asbestos exposure levels in the Korean GPJEM [15].

Non-occupational asbestos exposure was identified separately because individuals often had multiple exposure circumstances. Exposures unrelated to work included living in proximity to industrial or natural sources of airborne asbestos (environmental exposures), sharing a home with individuals occupationally exposed to asbestos (familial exposure), having experienced a fire or an earthquake that destroyed buildings in the neighborhood and living close to a public garbage dump. Detailed information was also collected from all subjects for several potential risk factors including sociodemographic data, smoking and exposure to second-hand smoke. Smoking was expressed in pack-years which was the number of years of smoking an average of 20 cigarettes per day.

### 2.3. Statistical Analysis

The questionnaire data, tumor histology and CT scan results collected from the study participants were entered into an electronic database using Epi-info software. Statistical analyses were performed using IBM SPSS Statistics 23 software (IMB Corporation, New York, NY, USA). The primary data analyses were based on logistic regression, adjusting all models for gender, age, ethnicity, education, house ownership, smoking and environmental asbestos exposure. Effects were reported as odds ratios and 95% confidence intervals and considered as significant at a level of 5%.

The combined effect of asbestos exposure and smoking on the risk of lung cancer was examined by cross-classification of the two variables with the categories “not exposed to asbestos” or “exposed less than 10 fiber-years (_A(−)_)” versus “exposed to asbestos 10 fiber-years or more (_A(+)_)” and, “not smoking or smoking less than 10 pack-years (_Sm(−)_)” versus “smoking 10 pack-years or more (_Sm(+)_)”. The interaction effect was evaluated using the synergy index (S) and multiplicativity (V) risk model. The synergy index measured the extent to which the risk ratio for smoking and asbestos together exceeds 1 and whether this is greater than the sum of the extent to which each of the risk ratios considered separately each exceeds 1. If S > 1, the additive interaction is said to be positive and if S < 1, the additive interaction is said to be negative. The synergy index was calculated using derived odd ratios (ORs) as follows [18]:
S = (OR_A(+)Sm(+)_ − 1)/((OR_A(+)Sm(−)_ − 1) + (OR_A(−)Sm(+)_ − 1)).

The multiplicativity index measured the extent to which, on the risk ratio scale, the effect of asbestos and smoking together exceeds the product of the effects of the two exposures considered separately. If V > 1, the multiplicative interaction is said to be positive. If V < 1, the interaction is said to be negative. The multiplicativity was calculated using derived ORs as follows [18]:
V = OR_A(+)Sm(+)_/(OR_A(+)Sm(−)_ × OR_A(−)Sm(+)_).

### 2.4. Ethics Approval

The Ethical Committee of Ludwig Maximilian University, Munich, Germany number 18-632 and the Ethics Committee of Persahabatan Hospital, Jakarta, Indonesia number 18/KEPK-RSUPP/03/2018 approved the study protocol. Subjects were informed about all aspects of the study, and they provided written consent for participation in the study.

## 3. Results

A total of 710 subjects were interviewed between May 2018 and August 2019. Among them, 336 subjects were eligible for cases, and 360 were eligible for the controls. Fourteen patients were excluded. Two subjects were not confirmed histologically of having lung cancer. Two subjects were excluded because of metastatic cancer from other organs to the lung, five subjects were suspected mesothelioma cases, three subjects were suspected asbestosis, and two subjects were suspected of interstitial lung diseases.

Table 1 presents the characteristics of the subjects. The mean age for cases was 58.1 years, and for control was 54.8 years. About 40% of the subjects reported more than 10 package-years of smoking. The proportions of males, of house owners and of smokers was higher in cases than in controls.

A total of 1095 work histories were collected from 710 individuals. 84 (12%) of the subjects, including 55 cases and 29 controls, reported that they worked at the industries or the occupations with asbestos exposures from the Korean GPJEM. Occupational asbestos exposure levels were between 0.03 fiber-years and 61.20 fiber-years with a mean exposure of 0.86 fiber-years in cases and 0.43 fiber-years in controls (*p* = 0.14). Only eight subjects were exposed to more than 25 fiber-years of asbestos. The most common occupation was technical workers in construction, the second was automobile mechanic and the rest were in other industries that handled asbestos-containing material (Table 2).

More than 40% of subjects reported having been exposed environmentally to asbestos roofing and a few to other environmental exposures (Table 3).

Table 4 shows the histological cell type of the lung cancer and the underlying diagnoses of the control group. The most frequent histological type was adenocarcinoma and the most frequent underlying non-malignant diagnosis was tuberculosis. No significant association has been found between asbestos exposure and the histological cell type within the case group nor between exposure and the underlying diagnosis within the control group.

The odds ratio of lung cancer in exposed subjects was 2.04-fold compared to unexposed subjects after adjusting for age, gender, homeownership, education and smoking (OR = 2.04, 95% CI = 1.21–3.42). It increased three-fold for subjects with cumulative asbestos exposure of 10 fiber-years or more (OR = 3.08, 95% CI = 1.01–9.46)). When the exposure was categorized into unexposed, exposed by less than 25 f-year and exposed 25 f-year or more, the ORs showed a dose–response relationship but did not achieve a 5% significance level. Concerning the duration, exposure to asbestos for more than 10 years doubled the chance of lung cancer (OR = 2.31, 95% CI = 1.26–4.26). The OR of a latency period between 10 to 30 years was 2.47 (95% CI = 1.10–5.55). No significant effect of environmental asbestos exposure was observed (Table 5).

Smoking significantly doubled the chance of getting lung cancer (OR = 1.88, 95% CI = 1.25–2.83). The combined effect of asbestos exposure and smoking in the development of lung cancer is shown in Table 6. After adjusting for age, gender, education and homeownership, subjects with cumulative asbestos exposure 10 fiber-years or more combined with smoking 10 pack-years or more had an 8.7-fold increased risk of lung cancer compared to the reference (OR = 8.70, 95% CI = 1.71–44.39). The synergy index for the association of smoking and asbestos exposure was 6.2 and the multiplicativity index was 3.4. Those results supported the positive additive and multiplicative interaction between smoking and asbestos.

## 4. Discussion

The present study proves our hypothesis that exposure to asbestos is associated with an increased risk of lung cancer among Indonesian workers. This association persisted after adjusting for age, sex, education, homeownership and smoking. The increased risk is consistent with a dose–response relationship. This study shows that asbestos exposure contributes to the lung cancer burden in Indonesia.

Unfortunately, Indonesia does not have national figures for lung cancer diagnosis. Our study was performed at the National Respiratory Referral Hospital in Jakarta, that is the national referral hospital for respiratory diseases and the most prominent center for lung cancer management in Indonesia, which allows us to assume that we obtained a representative sample of the lung cancer patients of Indonesia for the aim of our study.

We observed that the chance of getting lung cancer more than doubled among exposed subjects compared with unexposed subjects and it was comparable to the findings of Hardt et al. [19]. In their case-control study, those who were grouped as exposed subjects by DOM-JEM assessment had an odds ratio of 1.9 (95% CI = 1.3–2.7) compared to those with no exposure. Our study was similar to the study of Hardt et al. because both defined asbestos exposures using the job-exposure matrix from a neighboring country.

The dose–response relationship was explored in this study by employing fiber-years, categorized into unexposed, exposed to less than 10 fiber-years and exposed to 10 fiber-years or more. While the risk of lung cancer increased with an exposure of up to 10 fiber-years by 85%, we found an increase of 200% for subjects exposed to 10 fiber-years or more. It was somewhat lower than the effect reported by Yano et al. for high asbestos exposure in a study of asbestos textile workers in China (OR: 3.66, 95% CI 1.61 to 8.29), however the confidence interval of OR between this study and Yano’s study are in the range of adjacent intervals [20].

Other published studies reported lower ORs for the risk of asbestos exposure leading to lung cancer. Villeneuve et al., in a population-based case-control study in Canada, reported the OR 1.28 (95% CI: 1.02–1.61) for generally exposed subjects and when the exposure categorized into low and medium or high exposure the ORs was 1.17 (95% CI: 0.92–0.50) and 2.16 (95% CI: 1.21–3.88) respectively [21]. In a meta-analysis, Moon et al. concluded that the great variety of asbestos exposures between nations may arise from differences in culture, technology, legislation and attitude toward the risk [22]. The prevalence of smokers in different countries also strongly influenced the risk of lung cancer.

This present study found a significant association between the duration of asbestos exposure measured in years and the risk of lung cancer that supported the results from previous studies [19,23]. This association is slightly weaker as it does not take the intensity into account. However, given the limited information regarding actual asbestos exposure levels in many studies, the duration of exposure can be presumed to be a valid proxy measure of cumulative exposure.

The combined effect of smoking and asbestos exposure is of special interest in a country like Indonesia where smoking prevalence is very high. The combination of high exposure to asbestos and smoking increased the chances of lung cancer by 770%. The same effect was found by Ngamwong et al. in a meta-analysis [24]. The combined effect between smoking and asbestos exhibited a positive additive and multiplicative interaction. The strong interaction of both risk factors pointed out the need for lung cancer prevention efforts, especially in Indonesia.

Indonesia is one of the countries in Asia that has no comprehensive reporting of asbestos-related diseases despite its widespread use. Having no available exposure data made estimates of asbestos exposure challenging. In this situation, adopting the JEM from another Asian country as a reference to estimate the exposure was more reasonable than other methods such as exposure self-reports by study subjects or qualitative estimates of exposures. The General Population JEM from Korea was the best choice for several reasons. First, the Korean GPJEM grouped exposures based on international industrial classifications that linked to Indonesian classifications [15,25]. Second, asbestos processing companies and the asbestos-containing material used are similar in both countries; in fact, several asbestos industries were transferred from Korea to Indonesia [26]. The GPJEM brought more certainty and consistency to the asbestos exposure values for each industry and that led to a more precise classification of exposure.

However, differences in workplace conditions, culture, policy regarding asbestos management and the perception toward hazards between Indonesia and Korea could be a potential bias in our risk estimate. Yet, this would not refute the general results that asbestos exposure is associated with a significantly increased risk of lung cancer, which has also been confirmed by regarding the duration of exposure and is consistent with other similar studies in the epidemiological literature.

## 5. Conclusions

The findings of the present case-control study are consistent with other previous studies that supported the association of asbestos exposure and lung cancer. The disease risk is consistent with a dose-response relationship. It brought crucial new information regarding asbestos as the cause of lung cancer in Indonesia. Furthermore, there is a strong interaction of positive additive and multiplicative effects between asbestos and smoking that has to be taken into account in the prevention of lung cancer efforts in the future.

## Figures and Tables

**Table 1 ijerph-17-00591-t001:** Characteristics of subjects.

		Cases (n = 336)		Controls (n = 360)	
	Mean (SD)	No. of Subjects (%)	Mean (SD)	No. of Subjects (%)	*p*-Value *
**Mean Age**	58.19 (9.79)		54.68 (10.35)		0.00
Gender					
Female		125 (37.2)		165 (45.8)	0.02
Male		211 (62.8)		195 (54.3)	
**Ethnicity**					
Javanese		115 (34.2)		111 (30.8)	0.65
Sundanese		57 (17.0)		54 (15.0)	
Sumatran		63 (18.8)		68 (18.9)	
Sulawesi		7 (2.1)		10 (2.8)	
Kalimantan		0 (0)		2 (0.6)	
Papuan/East Indonesia		4 (1.2)		5 (1.4)	
Others		90 (26.8)		110 (30.6)	
**Education**					
Illiterate		10 (3.0)		10 (3.0)	0.09
Elementary		76 (22.6)		64 (17.8)	
Junior High School		31 (9.2)		40 (11.1)	
Senior High School		130 (38.7)		170 (47.4)	
Bachelor		78 (23.3)		71 (19.8)	
Postgraduate		11 (3,3)		5 (1.4)	
**Homeownership**					
Renting		48 (14.3)		69 (19.2)	0.01
Own house		234 (69.6)		195 (54.3)	
Family house		54 (16.1)		96 (26.7)	
**Smoking**					
0 to 10 pack-years		172 (51.2)		236 (65.6)	0.001
>10 to 40 pack-years		106 (31.5)		80 (22.2)	
>40 pack-years		58 (17.3)		44 (12.3)	

* The *t*-test was used for continuous variables and χ^2^ test for categorical variables.

**Table 2 ijerph-17-00591-t002:** Distribution of occupations with exposure to asbestos based on Indonesian Standard Industrial Classification (ISIC) 2015 and Korean Standard Industrial Classification (KSIC) Rev 9.

				Cases	Controls
ISIC	KSIC	Industrial Classification	Occupation	n	n
Handling Asbestos-Containing Product During Working					
410	41,112	Construction	Technical worker	33	19
4520	95,212	Car repair and maintenance	Automobile mechanic	9	4
28,160	29,169	Manufacture of lifting and transferring equipment	General machinery assembler	1	0
29,200	30,399	Four-wheeled or more vehicles carrosserie industry	Automobile paint mechanic	1	0
6110	6022	Telecommunication activities with cables	Technician	2	1
25,920	25,921	Industrial services for metal and non-metal processing	Heat treatment operator	2	1
25,952	25,924	Manufacture of other fabricated metal products	Machine operator	1	0
2392	23,229	Manufacture of other refractory ceramic products	Operator	1	0
33,151	95,119	Ship, boat and floating building repair services	Ship mechanic	2	0
3011	30,111	Ship, boat and floating building industries	Ship assembler	0	1
**Handling Asbestos-Containing Product in Manufacturing Process**					
23,955	23,994	Manufacture of construction material from asbestos	Machine operator	0	1
17,021	17,129	Manufacture of paper and corrugated paper board	Machine operator	1	0
20,131	20,302	Manufacture of synthetic resin and other plastic materials	Machine operator	0	1
22,230	22,250	Manufacture of plastic pipe and equipment	Machine operator	2	1

Exposure group in Korean GPJEM. ISIC: Indonesian Standard Industrial Classification 2015; KSIC: Korean Standard Industrial Classification rev.9.

**Table 3 ijerph-17-00591-t003:** Distribution of environmental exposure.

Environmental Exposure	Cases	Control
	N (%)	N (%)
Unexposed	187 (55.6)	182 (50.6)
Asbestos roof	144 (42.8)	168 (46.7)
Demolition	1 (0.3)	2 (0.6)
Earthquake	2 (0.6)	0 (0)
Fire	1 (0.3)	1 (0.3)
Combination of two or more environmental exposures	1 (0.3)	7 (1.9)

**Table 4 ijerph-17-00591-t004:** Histological cell type of case group and the underlying diagnosis of the control group according to occupational asbestos exposure.

	Number (%)	Occupational Exposure (%)	No Occupational Exposure (%)	*p*-Value *
Histological Cell Type of Cases				
Adenocarcinoma	188 (55.9)	34 (61.8)	154 (54.8)	0.255
Squamous cell	72 (21.3)	13 (23.6)	59 (20.8)	
Small cell	13 (3.8)	3 (5.5)	10 (3.5)	
Large cell	16 (4.7)	0 (0)	16 (5.7)	
Others	44 (13.3)	4 (7.3)	40 (14.5)	
Unidentified	3 (0.9)	1 (1.8)	2 (0.7)	
	336 (100)	55 (100)	281(100)	
**Underlying Diagnosis of Controls**				
Tuberculosis and other lung infections	197 (54.9)	19 (65.5)	178 (54.1)	0.856
Chronic lung diseases	62 (17)	5 (16.7)	57 (17.0)	
Mediastinal mass and other malignancies	40 (11.1)	1 (3.4)	39 (11.6)	
Other diseases	31 (8.6)	2 (6.7)	29 (8.8)	
No abnormality	30 (8.4)	2 (6.7)	28 (8.5)	
	360 (100)	29 (100)	331(100)	

* χ^2^ test used for categorical variables.

**Table 5 ijerph-17-00591-t005:** Odds ratio and adjusted odds ratio of the association between asbestos exposure and lung cancer.

	Cases (n = 336)	Controls (n = 360)	OR (95% CI)	Adjusted OR (95% CI) ^b^
	No of Subjects (%)	No of Subjects (%)		
**Occupational Asbestos Exposure**				
Unexposed	281 (83.6)	331 (91.9)	1	1
Ever exposed	55 (16.4)	29 (8.1)	2.23 (1.39–3.60)	2.04 (1.21–3.42)
**Categorical of Exposure by Fiber-Years**				
Unexposed	281 (83.6)	331 (92.5)	1	1
Exposed <10 fiber-years	44 (13.1)	24 (6.7)	2.16 (1.28–3.64)	1.85 (1.05–3.24)
Exposed ≥10 fiber-years	11 (3)	5 (0.8)	2.59 (0.89–7.55)	3.08 (1.01–9.46)
**Duration of Exposure**				
0 years	281 (83.6)	331 (91.9)	1	1
<10 years	13 (3.9)	10 (2.8)	1.53 (0.66–3.55)	1.34 (0.56–3.20)
≥10 years	42 (12.5)	19 (5)	2.60 (1.48–4.58)	2.31 (1.26–4.26)
**Latency Period from First Exposure**				
Never contact	281 (83.6)	331 (91.9)	1	1
10 to 30 years	22 (6.5)	10 (2.8)	2.59 (1.21–5.56)	2.47 (1.10–5.55)
>30 years	33 (9.8)	19 (4.7)	2.05 (1.14–3.68)	1.85 (0.98–3.50)
**Environmental Asbestos Exposure**				
Unexposed	187 (55.7)	181 (50.5)	1	1
Ever exposed	149 (44.3)	178 (49.6)	0.8 (0.60–1.09)	0.77 (0.56–1.05)

The OR was calculated using logistic regression. ^b^ adjusted for age, gender, homeownership and smoking.

**Table 6 ijerph-17-00591-t006:** Odds ratio and adjusted odds ratio of the association between the combination of occupational asbestos exposure and smoking with lung cancer.

	Cases (n = 336)	Controls (n = 360)	OR (95% CI)	Adjusted OR (95% CI) ^c^
	No of Subjects (%)	No of Subjects (%)		
_A(−) Sm(−)_	169 (50.3)	233 (64.7)	1	1
_A(+) Sm(−)_	3 (0.9)	3 (0.9)	1.38 (0.28–6.92)	1.38 (0.25–7.62)
_A(−) Sm(+)_	156 (46.4)	122 (33.9)	1.76 (1.30–2.40)	1.85 (1.20–2.85)
_A(+) Sm(+)_	8 (2.4)	2 (0.6)	5.52 (1.16–26.30)	8.70 (1.71–44.39)
S				6.2 ^d^
V				3.4 ^e^

The OR was calculated using logistic regression. ^c^ Adjusted for age, gender and homeownership; _A(−)_: Unexposed and exposed to asbestos less than 10 fiber-years; _A(+)_: Exposed to asbestos 10 fiber-years or more; _Sm(−)_: Smoking 0–10 pack-years; _Sm(+)_: Smoking 10 pack-years or more; ^d^ S = (OR_A(+)Sm(+)_ − 1)/[(OR_A(+)Sm(−)_ − 1) + (OR_A(−)Sm(+)_ − 1)]; ^e^ V = OR_A(+)Sm(+)/[_OR_A(+)Sm(−)_ × OR_A(−)Sm(+)_].

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
