# Peer review of "Asbestos-Related Lung Cancer: A Hospital-Based Case-Control Study in Indonesia"

_ijerph, 2020, doi:10.3390/ijerph17020591_

Round 1

Reviewer 1 Report

Asbestos-Related Lung Cancer: A Hospital-Based Case Control Study in Indonesia

In total, 710 subjects, including 336 lung cancers and 360 controls, were interviewed for their work and residential environment. Asbestos exposure was assessed based on Job-Exposure Matrix developed for Korean industry, as the exposure measurements were not available in Indonesia. After controlling covariates, the results show elevated odd ratio of lung cancer from asbestos exposure.

This study adds important evidence of asbestos hazardous effects in developing countries, especially under the circumstances of controversy of chrysotile from Russia in Asian countries. However, the study needs to address the following comments before the results can be interpreted appropriately.

The study does not mention about the lung cancer diagnosis, except a short sentence of two deleted cases of not histologically confirmed. Authors need to give details of the final diagnosis of lung cancer groups, including the histologic diagnosis. If possible, authors need to explain about the national figures of lung cancer diagnosis, so that readers can understand how the studied group can represent the whole lung cancer patient population in Indonesia, in terms of the size, histologic types, etc. Authors also need to divide the case group into histologic categories, when the association with asbestos exposure was examined. Authors need to give details of the underlying diagnosis of the control group. How the controls were selected is very crucial for the validity of the study, and the process and criteria for inclusion or exclusion should be specified more clearly, and the results should be shown according to the procedures and criteria. The industry structure of Indonesia and its classification should be compared to those of Korea, in order to assess the appropriateness of using the exposure classification of different industries.

Reviewer 2 Report

This study presented a research on correlation between occupational asbestos exposure and risk of lung cancer development.

The Introduction is comprehensive and provides a solid background for a reader.

The Methods section is presented properly and is detailed enough for the research to be repeated.

In the Results section authors have addressed all concerns very well. I really like the fact that the author estimated not only occupational asbestos exposure but its co-existing with other risk factors for lung cancer development, and also non-occupational exposure. The tables and figures are comprehensive and comfortable to read.

The Discussion section is well-structured. The authors take their results critically.

The only grammatical mistake I found is placed on pg. 7 line 196: space is not required before closing parenthesis.

The paper is well-written and easy to read, the authors have addressed all concerns.

Author Response

Thank you very much for your review and encouraging comments, which we appreciate them.

We have made a correction regarding your comments.